# Inflammasome genes polymorphisms may influence the development of hepatitis C in the Amazonas, Brazil

**Diana Mota Toro**[1], **Rajendranath Ramasawmy**[1,2,3,4,5], **Pedro Vieira Silva Neto**[1], **Grenda Leite Pereira**[1], **Priscila Santos Sarmento**[1], **Hanna Lara Silva Negreiros Dray**[6], **Keyla Santos Sousa**[6,7], **Juliana Santos Affonso**[6,7], **Jéssica Albuquerque Silva**[6], **Nadja Pinto Garcia**[6], **Marilú Victória Barbieri**[2,3], **Flamir Silva Victória**[2,3], **Eduardo Antônio Donadi**[8], **Allyson Guimarães Costa**[1,2,3,5,7,6,9] *, **Mauricio Morishi Ogusku**[1,5,10], **Aya Sadahiro**[1,5], **Andréa Monteiro Tarragô**[1,5,6,7], **Adriana Malheiro**[1,2,5,6,7] *

1 Programa de Pós-Graduação em Imunologia Básica e Aplicada, Universidade Federal do Amazonas, Manaus, Amazonas, Brazil, 2 Programa de Pós-Graduação em Medicina Tropical, Universidade do Estado do Amazonas, Manaus, Amazonas, Brazil, 3 Instituto de Pesquisa Clínica Carlos Borborema, Fundação de Medicina Tropical Dr. Heitor Vieira Dourado, Manaus, Amazonas, Brazil, 4 Faculdade de Medicina, Universidade Nilton Lins, Manaus, Amazonas, Brazil, 5 Genomic Health Surveillance Network: Optimization os Assistance and Research in The State of Amazonas -REGESAM, Manaus, Amazonas, Brazil, 6 Diretoria de Ensino e Pesquisa, Fundação Hospitalar de Hematologia e Hemoterapia do Amazonas, Manaus, Amazonas, Brazil, 7 Programa de Pós-Graduação em Ciências Aplicadas à Hematologia, Universidade do Estado do Amazonas, Manaus, Amazonas, Brazil, 8 Programa de Pós-graduação em Imunologia Básica e Aplicada, Faculdade de Medicina da Universidade de São Paulo, Ribeirão Preto, São Paulo, Brazil, 9 Escola de Enfermagem de Manaus, Universidade Federal do Amazonas, Manaus, Amazonas, Brazil, 10 Laboratório de Micobacteriologia, Instituto Nacional de Pesquisas da Amazônia, Manaus, Amazonas, Brazil

* allyson.gui.costa@gmail.com (AGC); malheiroadriana@yahoo.com.br (AM)

**Data Availability Statement:** Due to ethical restrictions regarding patient privacy, data are available upon request. Data are available upon request from the Ethics Committee for researchers

## Abstract

Hepatitis C is considered a major public health problem caused by the hepatitis C virus (HCV). Viral infections are known to induce production of IL1β through the signaling pathway of inflammasomes. Emerging evidences suggest that Inflammasome genes may influence the immune response against HCV as the host genetic background may contribute to the balance between acute and chronic inflammation. We investigated in 151 patients with chronic hepatitis C and 206 healthy blood donors' individuals (HD). Polymorphisms in the *IL1B* and *IL18* genes were genotyped by PCR-RFLP, while *NLRP3*, *CARD8*, *CTSB* and *AIM2* by RT- PCR. Serum assay of IL-1β cytokine was performed by ELISA. 84 patients presented mild fibrosis (<F2) and 67 advanced fibrosis ($\geq$ F2). Among the HD individuals the *NLRP3*-rs10754558 C/C genotype correlated with higher IL-1β levels compared to the G/G genotype. Similar pattern was observed in patients with hepatitis C, mean circulating IL-1β levels were 21,96 ± 4.5 and 10,62 ± 3.3pg/mL among the C/C and G/G genotypes, respectively. This pattern holds even after stratification of the patients into mild fibrosis and advanced fibrosis, demonstrating that the *NLRP3*-rs10754558 or another polymorphism in linkage disequilibrium with it possibly has an influence on the processing of pro-IL-1β. Notably, higher levels of IL-1β (Mann–Whitney test, p<0.0001) were observed among patients (mean ± SEM: 19,24 ±3.pg/mL) when compared with controls (mean ± SEM: 11,80 ±1.0pg/mL). Gene-gene interaction showed that individuals heterogyzotes for both *CARD8*-

from Fundação de Medicina Tropical Dr. Heitor Vieira Dourado (CEP/FMT-HVD - cep@fmt.am.gov.br), for researchers who meet the criteria for access to confidential data. Additional requests for the data may be sent to the corresponding author or coauthors Allyson G. Costa (allyson.gui.costa@gmail.com); Adriana Malheiro (malheiroadriana@yahoo.com.br); Andréa Monteiro Tarragô (andrea_s_monteiro@hotmail.com); Rajendranath Ramasawmy (ramasawm@gmail.com).

**Funding:** This work was funded by Fundação de Amparo à Pesquisa do Estado do Amazonas (FAPEAM) (Pró-Estado Program - #002/2008, #007/2018 and #005/2019, PAMEQ Program - #004/2019, PAPAC Program - #005/2019 and PECTI-AM/SAÚDE Program #004/2020), Rede Genômica de Vigilância em Saúde do Estado do Amazonas (REGESAM), Conselho Nacional de Desenvolvimento Científico e Tecnológico (CNPq) (Universal Program - #407818/2016), Coordenação de Aperfeiçoamento de Pessoal de Nível Superior (CAPES) and Brazilian Ministry of Health.

**Competing interests:** The authors have declared that no competing interests exist.

rs2009373 and *IL1B*-rs16944 are less prone to hepatitis C development ($p_{adj}$ = 0.039). Similarly, herozygote carriers for *CTSB*-rs1692816 and *AIM2*-rs1103577 ($p_{adj}$ = 0.008) or for *IL18*-rs187238 and *NLRP3*-rs10754558 ($p_{adj}$ = 0.005), have less chances to the development of hepatitis C. However, between subgroups of <F2 and ≥F2, individuals homozygous for the T allele of *CARD8*-rs2009373 and heterozygous for IL18-rs187238 ($p_{adj}$ = 0.028), have mild form of fibrosis.

## Introduction

Hepatitis C is an inflammatory liver disease caused by the hepatitis C virus (HCV). Nearly 71 million individuals are estimated to be infected worldwide and around 70% will progress to chronic form of the disease and may develop liver fibrosis, cirrhosis and hepatocellular carcinoma [1, 2].

Persistent inflammatory response activates immunoregulatory mechanisms and stimulates the production of TGF-β and IL-10, regulatory cytokines. TGF-β1 is a potent activator of hepatic stellate cell (HSCs) trans differentiation in myofibroblasts that stimulate the mechanisms of hepatic fibrosis [3–8]. IL-1β is also capable of stimulating fibrosis by activating HSCs cells and contributing to the development and maintenance of fibrosis in the liver [9].

IL-1β induces COX2, nitric oxide and TNF-α, and also modulates various cellular processes during chronic HCV infection [10]. Persistent hepatic macrophage IL-1β production in patients with chronic HCV attracts immune cells to the liver, enhancing inflammation [11, 12]. Exacerbated proinflammatory cytokine production contributes to tissue damage and development of autoinflammatory diseases [13].

Processing of pro-IL1β and pro-IL-18 proinflammatory cytokines into their bioactive forms requires the formation of molecular complexes called inflammasomes [14]. NLRP3 inflammasome is extensively studied and its gene expression is positively regulated by the transcription factor NF-kB [15–19]. CARD8 negatively regulates inflammasome activity, inhibiting activation of transcription factor NF-kB and regulating activation of inflammatory caspases [20, 21]. CARD8 regulates bioactive IL-1β secretion through inhibitory interaction with NLRP3 caspase-1 [20] and / or acting through inhibitory mechanisms in pro-IL-1β formation via the NF-kB pathway [20–23].

Growing evidence suggests that inflammasome genes may influence the immune response against HCV. Since nearly two-third of HCV-infected individuals progress to chronic form of the disease while the rest can eliminate the virus or remain asymptomatic, this suggests that the host genetic background may play an important role in the clinical outcome of HCV infection. In the present study, we analyzed whether polymorphisms in inflammasomes genes *IL1B* rs16944, *IL18* rs187238, *NLRP3* rs10754558, *CARD8* rs2009373, *CTSB* rs1692816 and *AIM2* rs1103577 are associated with susceptibility to HCV infection and liver fibrosis in hepatitis C patients in the Amazon population.

## Material and methods

### Ethics approval

This study was reviewed by the Ethics Committee of the Fundação Hospitalar de Hematologia e Hemoterapia do Amazonas (HEMOAM) and granted under the file (CAAE 49652815.8.0000.0009 and CAAE 0024.112.000–10). All of the study participants provided a

signed written informed consent form prior to the enrollment in the study, according to Declaration of Helsinki and Resolution 466/12 of the Brazilian National Health Council for research involving human subjects. All patients were treated following the specifications of the Brazilian Ministry of Health [24].

## Samples and clinical data

The study was performed at the Fundação de Medicina Tropical Dr. Heitor Vieira Dourado (FMT-HVD) in Manaus, the capital city of the Amazonas State, during 2016–2017. The study population was described elsewhere [25]. Briefly, 206 Healthy blood donors (HD) were randomly selected from HEMOAM and 151 HCV infected patients were recruited at the FMT-HVD. Among the HCV infected patients, 84 and 67 had < F2 and ≥F2, respectively.

## Sample collection and genomic DNA extraction

Sample collection and genomic DNA extraction are also described elsewhere [25]. Briefly, genomic DNA was extracted from peripheral blood samples using the QIAamp DNA Blood Mini Kit (QIAGEN, Chatsworth, CA, USA) according to the manufacturer's instructions.

## Molecular analysis of polymorphisms

6 SNPs in inflammasome genes (CARD8, CTSB, NLRP3, AIM2, IL1B, IL18) were selected according to previously reported association studies and public databases (NCBI dbSNP and Human genome GRCh37.p13).

The NLRP3 rs10754558, CTSB rs1692816, CARD8 rs2009373 and AIM2 rs1103577 were identified by Real Time PCR. Whereas IL1B rs16944 and IL18 rs187238 of were identified by PCR-RFLP.

Real-time PCR reactions were performed with 3.5μL of ultrapure water, 5μL of Master Mix genotyping (1x), 0.5μL of TaqMan® assay (20x), containing 36μM of each primer and 8μM of each TaqMan® probe, 1μL genomic DNA, with final volume of 10μL (S1 Table). Applied Biosystems QuantStudio™ 3 Thermocycler by Thermofisher Scientific was used for amplification of fragments of interest, with the following cycling parameters: 95oC for 10 minutes for activation, 50 cycles at 92oC for 15 seconds for denaturation and 50 cycles at 60oC for 90 seconds for annealing and extension.

PCR-RFLP were performed in the Eppendorf Mastercycler ep in a final volume of 25uL consisting of ~20ng genomic DNA, 2U Platinum™Taq DNA polymerase (Thermo Fisher Scientific), 2.5μL10x buffer (100 mmol/L Tris-HCl (pH 8.3) and 500 mmol/L KCl), 1μL MgCl2 (1.5 mmol/L), 1μL dNTPs (40 mmol/L), and 0.25 pmol/L each of forward and reverse primer. A total of 10μL of PCR product was digested with 5U of restriction enzyme (AvaI for IL1B rs16944 and MboII for IL18 rs187238) with their specified buffer. The restrictions enzymes are from New England Biolabs, Ipswich, MA, USA. The primers, PCR cycling conditions, and restriction endonucleases are shown in S2 Table. After digestion with the restriction enzymes, the fragments generated were size-separated by electrophoresis in a 2% - 4% agarose gel stained with GelRed™ Nucleic Acid Gel Stain (Biotium, Hayward, CA, USA), and visualized with the UV light Gel Doc™ XR +System (Bio-Rad Corporation, Hercules, CA, USA) with a photo documentation system.

## Serum cytokine assay

Serum concentrations of IL-1β was measured by the Enzyme-Linked Immunosorbent Assay (ELISA) sandwich immunoassay. Dosages were made from plasma of subjects in the Hepatitis

C patient group and controls. We use a commercial BD® Biosciences (San Jose, CA, USA) human BDTM OptEIA® Set II kit for cytokine IL-1β.

## Genotype association test and statistical analyzes

Demographic, clinical and laboratory data were presented in tables and graphs, prepared using the Excel program (Microsoft Corporation). Categorical variables are expressed as absolute value (n) and relative frequency (%). Statistical analysis between independent groups was performed by Chi-square or Fisher's exact test for categorical variables and Student's t-test, Mann-Whitney test or ANOVA for continuous variables. Logistic regression was performed to identify risk factors associated with the specific outcome (advanced fibrosis), as well as a description of their "odds ratio" and 95% confidence interval. A significant threshold of $p < 0.05$ was adopted.

R software version 3.2.2 (www.r-project.org) was used to perform genotypes association and inheritance modelling (package SNP assoc version 1.9–2). Data were adjusted for sex and age. Polymorphisms were evaluated for Hardy-Weinberg equilibrium (HWE) and the association analysis with risk or protection for infection and fibrosis was performed for Codominant, dominant, recessive and Overdominate genetic inheritance models. The best genetic inheritance model was chosen according to the lowest value of Akaike Information Criterion (AIC).

## Results

### Basic characteristics of the study population

151 patients with chronic HCV infection and 206 healthy controls were included in the study. Among the HCV patients, 83 (55%) were males and 68 (45%) females, with a mean age of 57.8 ± SD 11.2 years. Of the HD, 144 (70%) were males and 62 (30%) females, with a mean age of 32 ± SD 10.8 years. The distribution of sexes was significantly different between the two studied groups ($p < 0.05$). Males were predominant in both groups ($p = 0,003$).

49 patients were treatment naïve. 84 patients (55.6%) had mild fibrosis (<F2) and 67 (44.4%) (≥F2) advanced fibrosis according to APRI (AST to Platelet Ratio Index) and FIB4 (Fibrosis-4) indices. Patients with advanced fibrosis (≥F2) had a higher mean age (61.03 ± 9.3), with $p = 0.001$. About viral genotypes, we detected 103 (68%) genotype 1, 32 (21%) genotype 3, 10 (7%) genotype 2 and one patient had the hepatitis C genotype 4.

### Frequency of genotypes and alleles

The genotypes and allele frequencies of all the studied SNPs are shown in Table 1. No significant deviation from the Hardy-Weinberg equilibrium was observed in any of the studied SNPs among the HD and the patients with HCV.

The frequency distribution of the *IL1B* rs16944 genotypes T/T, T/C, and C/C were 29%, 46%, and 25% among the patients and 33%, 50%, and 17%, among the HD, respectively ($p = 0.2$). Comparison of *IL1B* C/T rs16944 genotypes between patients with HCV infection and HD showed that carriers homozygote for the C allele seem to be susceptible to HCV infection after adjusting for age $p_{adj} = 0.054$, [OR = 1.68 (95% CI = 0.99–2.85)] as shown in Table 1. The other SNPs did show any significant association with susceptibility or resistance to HCV infection (Table 1).

We stratified the patients with HCV into mild fibrosis (<F2) and advanced fibrosis (≥F2) patients to look for if any of the SNP may indicate severity of the disease, as shown in Table 2. We observed that individual's homozygote for the A allele of *CTSB* rs1692816 SNP have mild fibrosis compared to individuals with the C allele $p = 0.044$ [OR = 0.46 (95% CI = 0.21–1.00)].

**Table 1. Relation of allele and genotype frequencies of SNPs with the hepatites C susceptibility.**

| Genotype and Allele | HD | Patients | p value | OR (95% CI) | p value | OR (95% CI) | $p_{adj1}$ | OR (95% CI) | $p_{adj2}$ | Comparisons |
|---|---|---|---|---|---|---|---|---|---|---|
| **IL1β rs16944** | (n = 203) | (n = 151) | | | | | | | | |
| T/T | 66 (33%) | 44 (29%) | 0.242 | (R) 1.56 (0.93–2.62) | 0.094 | (R) 1.68 (0.99–2.85) | 0.054 | (D) 1.75 (0.83–3.68) | 0.138 | T/Tvs C/T-C/C |
| C/T | 102 (50%) | 70 (46%) | | | | | | | | T/T-C/T vs C/C |
| C/C | 35 (17%) | 37 (25%) | | | | | | | | T/T-C/C vs C/T |
| T | 234 (58%) | 158 (52%) | 0.159 | | | | | | | |
| C | 172 (42%) | 144 (48%) | | | | | | | | |
| **IL18 rs187238** | (n = 202) | (n = 149) | | | | | | | | |
| C/C | 22 (11%) | 18 (12%) | 0.727 | (O) 84 (0.55–1.29) | 0.428 | (O) 0.86 (0.56–1.32) | 0.491 | (D) 0.72 (0.36–1.46) | 0.363 | G/G vs G/C-C/C |
| C/G | 94 (46%) | 63 (42%) | | | | | | | | G/G-G/C vs C/C |
| G/G | 86 (43%) | 68 (46%) | | | | | | | | G/G–C/C vs G/C |
| C | 138 (34%) | 99 (33%) | 0.795 | | | | | | | |
| G | 266 (66%) | 199 (67%) | | | | | | | | |
| **CARD8 rs2009373** | (n = 202) | (n = 151) | | | | | | | | |
| T/T | 43 (21%) | 28 (19%) | 0.418 | (D) 0.74 (0.47–1.17) | 0.194 | (D) 0.74 (0.46–1.18) | 0.201 | (R) 0.46 (0.18–1.16) | 0.098 | C/C vs C/T-T/T |
| C/T | 105 (52%) | 73 (48%) | | | | | | | | C/CC/T vs T/T |
| C/C | 54 (27%) | 50 (33%) | | | | | | | | C/C-T/T vs C/T |
| T | 191 (47%) | 129 (43%) | 0.228 | | | | | | | |
| C | 213 (53%) | 173 (57%) | | | | | | | | |
| **CTSB rs1692816** | (n = 202) | (n = 151) | | | | | | | | |
| C/C | 56 (28%) | 45 (30%) | 0.472 | (O) 0.85 (0.55–1.29) | 0.439 | (O) 0.84 (0.55–1.29) | 0.420 | (O) 0.52 (0.25–1.08) | 0.772 | C/C vs A/C-A/A |
| A/C | 98 (49%) | 67 (44%) | | | | | | | | C/C-A/C vs A/A |
| A/A | 48 (24%) | 39 (26%) | | | | | | | | C/C-A/A vs A/C |
| C | 210 (52%) | 157 (52%) | 0.998 | | | | | | | |
| A | 194 (48%) | 145 (48%) | | | | | | | | |
| **NLRP3 rs10754558** | (n = 206) | (n = 151) | | | | | | | | |
| G/G | 13 (6%) | 15 (10%) | 0.450 | (R) 1.64 (0.75–3.55) | 0.211 | (R) 1.71 (0.78–3.74) | 0.179 | (R) 0.55 (0.16–1.91) | 0.352 | C/C vs C/G-G/G |
| G/C | 74 (36%) | 53 (35%) | | | | | | | | C/C–C/G vs G/G |
| C/C | 119 (58%) | 83 (55%) | | | | | | | | C/C-G/G vs C/G |
| G | 100 (24%) | 83 (27%) | 0.331 | | | | | | | |
| C | 312 (76%) | 219 (73%) | | | | | | | | |
| **AIM2 rs1103577** | (n = 200) | (n = 151) | | | | | | | | |
| T/T | 39 (20%) | 29 (19%) | 0.306 | (D) 71 (0.45–1.12) | 0.140 | (D) 0.69 (0.43–1.10) | 0.117 | (D) 0.75 (0.35–1.61) | 0.461 | C/C vs C/T–T/T |
| T/C | 108 (54%) | 71 (47%) | | | | | | | | C/C-C/T vs T/T |
| C/C | 53 (27%) | 51 (34%) | | | | | | | | C/C-T/T vs C/T |
| T | 186 (47%) | 129 (43%) | 0.318 | | | | | | | |
| C | 214 (54%) | 173 (57%) | | | | | | | | |

HD: Healthy blood donors. OR: odds ratio. 95% CI: 95% confidence interval. $Pa_{dj1}$: p value adjusted by sex. $P_{adj2}$: p value adjusted by sex and age. The best genetic inheritance model was adopted for each SNP, according to the Akaike Information Criterion (AIC), where C = codominant, D = dominant, R = recessive, O = overdominant. The statistical analyses were conducted using the chi-square test. p < 0.05 is considered significant. Genetic models: Codominant: comparison of homozygote frequency for polymorphic allele, with heterozygote and homozygote for wild allele, simultaneously expressing both alleles. Dominant: Homozygote frequency comparison for wild allele with heterozygote + homozygote for polymorphic allele. Recessive: Comparison of the frequency of Homozygote for wild + heterozygous allele with homozygous for polymorphic allele. Overdominat: Comparison of frequencies Homozygote for wild allele + homozygote for polymorphic allele with heterozygote, which evaluates the heterozygous genotype.

However, when correcting for gender and age, individual's homozygote for the A allele still have 50% less chances of developing advanced fibrosis p = 0.091; [OR = 0.50 (95% CI = 0.22–1.13)].

**Table 2. Relation of allele and genotype frequencies of SNPs with the fibrosis susceptibility.**

| Genotype and Allele | <F2 | ≥F2 | p value | OR (95% CI) | p value | OR (95% CI) | $p_{adj1}$ | OR (95% CI) | $p_{adj2}$ | Comparisons |
|---|---|---|---|---|---|---|---|---|---|---|
| *IL1β* rs16944 | (n = 84) | (n = 67) | | | | | | | | |
| T/T | 20 (24%) | 24 (36%) | 0.498 | (O) 0.58 (0.30–1.11) | 0.095 | (D) 0.54 (0.27–1.11) | 0.091 | (D) 0.51 (0.24–1.08) | 0.076 | T/Tvs C/T-C/C |
| C/T | 44 (52%) | 26 (39%) | | | | | | | | T/T-C/T vs C/C |
| C/C | 20 (24%) | 17 (25%) | | | | | | | | T/T-C/CvsC/T |
| T | 84 (50%) | 74 (55%) | 0.665 | | | | | | | |
| C | 84 (50%) | 60 (45%) | | | | | | | | |
| *IL18* rs187238 | (n = 82) | (n = 67) | | | | | | | | |
| C/C | 9 (11%) | 9 (13%) | 0.720 | (O) 0.62 (0.32–1.19) | 0.147 | (O) 0.61 (0.32–1.19) | 0.143 | (O) 0.53 (0.26–1.08) | 0.075 | G/G vs G/C-C/C |
| C/G | 39 (48%) | 24 (36%) | | | | | | | | G/G-G/C vs C/C |
| G/G | 34 (41%) | 34 (51%) | | | | | | | | G/G–C/C vs G/C |
| C | 57 (35%) | 42 (31%) | 0.824 | | | | | | | |
| G | 107 (65%) | 92 (69%) | | | | | | | | |
| *CARD8* rs2009373 | (n = 84) | (n = 67) | | | | | | | | |
| T/T | 15 (18%) | 13 (19%) | 0.492 | (D) 1.90 (0.94–3.84) | 0.069 | (D) 1.88 (0.93–3.80) | 0.076 | (O) 1.72 (0.87–3.39) | 0.115 | C/C vs C/T-T/T |
| C/T | 36 (43%) | 37 (55%) | | | | | | | | C/C-C/T vs T/T |
| C/C | 33 (39%) | 17 (25%) | | | | | | | | C/C-T/T vs C/T |
| T | 66 (39%) | 63 (47%) | 0.402 | | | | | | | |
| C | 102 (61%) | 71 (53%) | | | | | | | | |
| *CTSB* rs1692816 | (n = 84) | (n = 67) | | | | | | | | |
| C/C | 24 (29%) | 21 (31%) | 0.389 | (R) 0.46 (0.21–1.00) | *0.044* | (R) 0.46 (0.21–1.00) | *0.045* | (R) 0.50 (0.22–1.13) | 0.091 | C/C vs A/C-A/A |
| A/C | 33 (39%) | 34 (51%) | | | | | | | | C/C-A/C vs A/A |
| A/A | 27 (32%) | 12 (18%) | | | | | | | | C/C-A/A vs A/C |
| C | 81 (48%) | 76 (57%) | 0.339 | | | | | | | |
| A | 87 (52%) | 58 (43%) | | | | | | | | |
| *NLRP3* rs10754558 | (n = 51) | (n = 67) | | | | | | | | |
| G/G | 4 (5%) | 11 (16%) | 0.072 | (C) 0.53 (0.26–1.09) | *0.012* | 0.52 (0.25–1.08) | *0.012* | (C) 0.47 (0.22–1.01) | *0.016* | C/C vs C/G-G/G |
| G/C | 36 (43%) | 17 (25%) | | | | | | | | C/C–C/G vs G/G |
| C/C | 44 (52%) | 39 (58%) | | | | | | | | C/C-G/G vs C/G |
| G | 44 (26%) | 39 (29%) | 0.853 | | | | | | | |
| C | 124 (74%) | 95 (71%) | | | | | | | | |
| *AIM2* rs1103577 | (n = 84) | (n = 67) | | | | | | | | |
| T/T | 14 (17%) | 15 (22%) | 0.909 | (R) 1.44 (0.64–3.25) | 0.376 | (R) 1.46 (0.65–3.29) | 0.363 | (R) 1.31 (0.56–3.08) | 0.534 | C/C vs C/T–T/T |
| C/T | 42 (50%) | 29 (43%) | | | | | | | | C/C-C/T vs T/T |
| C/C | 28 (33%) | 23 (34%) | | | | | | | | C/C-T/T vs C/T |
| T | 70 (42%) | 59 (44%) | 0.918 | | | | | | | |
| C | 98 (58%) | 75 (56%) | | | | | | | | |

Fibrosis degree was assessed using METAVIR score. The statistical analyses were conducted using the chi-square test. $p < 0.05$ is considered significant. OR: odds ratio. 95% CI: 95% confidence interval. $p_{Adj1}$: $p$ value adjusted by sex. $p_{Adj2}$: $p$ value adjusted by sex and age. The best genetic inheritance model was adopted for each SNP, according to the Akaike Information Criterion (AIC), where C = codominant, D = dominant, R = recessive, O = overdominant. Genetic models: Codominant: comparison of homozygote frequency for polymorphic allele, with heterozygote and homozygote for wild allele, simultaneously expressing both alleles. Dominant: Homozygote frequency comparison for wild allele with heterozygote + homozygote for polymorphic allele. Recessive: Comparison of the frequency of Homozygote for wild + heterozygous allele with homozygous for polymorphic allele. Overdominat: Comparison of frequencies Homozygote for wild allele + homozygote for polymorphic allele with heterozygote, which evaluates the heterozygous genotype.

**Table 3. Gene-gene interaction between healthy blood donors (n = 202) and patients with hepatitis C (n = 151).**

| Gene-gene interaction | *p* value | Median | *p* value adjusted | Median | Association with |
|---|---|---|---|---|---|
| *CARD8* rs2009373 vs *IL1B* rs16944 | | | | | |
| CARD8 C/T: C/T *IL1B* | *0.032* | -0.907 | *0.039* | -0.104 | Protection |
| *CARD8* rs2009373 vs *IL18* rs187238 | | | | | |
| CARD8 C/T: G/C IL18 | *0.056* | -0.981 | *0.059* | -0.104 | - |
| *CTSB* rs1692816 vs *AIM2* rs1103577 | | | | | |
| CTSB A/C: C/T *AIM2* | *0.002* | -0.865 | *0.008* | -0.089 | Protection |
| *IL18* rs187238 vs *NLRP3* rs10754558 | | | | | |
| IL18 G/C: C/G NLRP3 | *0.092* | -0.913 | *0.005* | -0.098 | Protection |

Median: Median value (Average of minimum and maximum residues) generated by the statistical program R is the parameter that determines whether it is a risk or protection factor. Median values greater than 1 are associated with risk and less than 1 with protection; Adjusted *p* value: *p* value adjusted by gender and age. p < 0.05 is considered significant.

## Gene-gene interaction of the SNPs between healthy blood donors and patients with hepatitis C

To evaluate whether polymorphisms in different genes could have a combined effect and influence susceptibility to HCV infection and/or development of liver fibrosis, we performed a gene-gene interaction analysis. Statistically significant results are reported in Tables 3 and 4.

Gene-gene interaction showed that individuals heterozygotes for both *CARD8* rs2009373 and *IL1B* rs16944 are less prone to hepatitis C development ($p_{adj}$ = 0.039). Similarly, heterozygote carriers for *CTSB* rs1692816 and *AIM2* rs1103577 ($p_{adj}$ = 0.008) or for *IL18* rs187238 and *NLRP3* rs10754558 ($p_{adj}$ = 0.005), have less chances to the development of hepatitis C (Table 3).

## Gene-gene interaction of the SNPs between mild fibrosis (<F2) and advanced fibrosis (≥F2)

In the analysis of gene interaction between subgroups of mild fibrosis (<F2) and advanced fibrosis (≥F2), individuals homozygous for the T allele of *CARD8* rs2009373 and heterozygous for *IL18* rs187238 ($p_{adj}$ = 0.028), have mild form of fibrosis (Table 4).

## Influence of the SNPs on IL1β serum concentration

Of all the polymorphisms studied (S1 and S2 Figs), only *NLRP3* rs10754558 showed an influence on the serum cytokine concentration of IL-1β (Fig 1).

**Table 4. Gene-gene interaction between subgroups of patients with hepatitis C (<F2 n = 84 and ≥F2 n = 67) and relation of SNPs with the fibrosis.**

| Gene-gene interaction | *p* value | Median | *p* value adjusted | Median | Association |
|---|---|---|---|---|---|
| *CARD8* rs2009373 vs *IL18* rs187238 | | | | | |
| CARD8 T/T: G/C IL18 | *0.051* | -0.633 | *0.028* | -0.522 | Protection |
| *CTSB* rs1692816 vs *AIM2* rs1103577 | | | | | |
| CTSB A/A: C/T *AIM2* | *0.060* | -0.417 | *0.052* | -0.413 | Protection |

Median: Median value (Average of minimum and maximum residues) generated by the statistical program R is the parameter that determines whether it is a risk or protection factor. Median values greater than 1 are associated with risk and less than 1 with protection; Adjusted p value: p value adjusted by gender and age. Fibrosis degree was assessed using METAVIR score. p < 0.05 is considered significant.

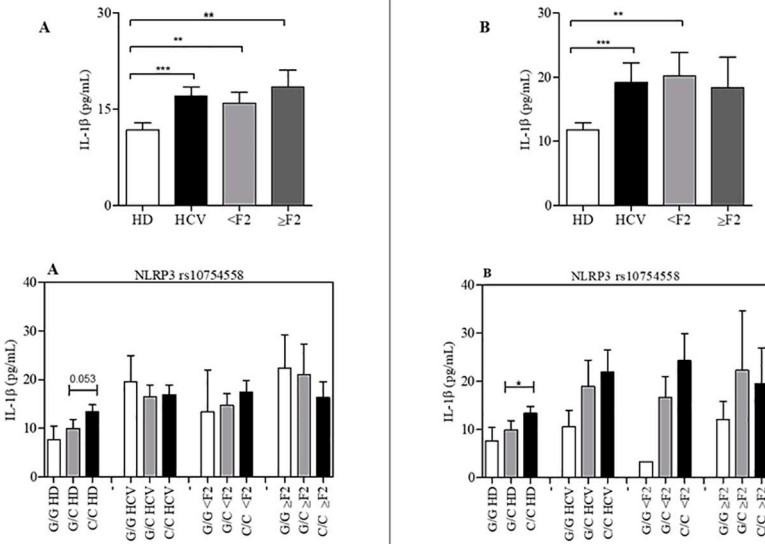

**Fig 1. Serum concentration of IL-1β cytokine in patients with hepatitis C.** Treated (A) and untreated (B), stratified according to the degree of hepatic fibrosis, analyzed as a function of the *NLRP3* rs10754558 polymorphisms. Data are expressed as mean ± standard deviation of circulating concentration (pg/mL) of IL-1β cytokine. Statistical analyzes were performed by ANOVA (nonparametric analysis of variance), with Kruskal-Wallis test, followed Dunn's post-test to compare pairs.

Among the HD individuals the *NLRP3* rs10754558 C/C genotype correlated with higher IL-1β levels (mean ± SD: 13.39 ± 1.4 pg/mL) compared to the G/G genotype (mean ± SD: 7,69 ± 2.0 pg/mL). Similar pattern was observed in patients with hepatitis C, mean circulating IL-1β levels were 21.96 ± 4.5 and 10.62 ± 3.3 pg/mL among the C/C and G/G genotypes, respectively. This pattern holds even after stratification of the patients into mild fibrosis and advanced fibrosis, demonstrating that the *NLRP3* rs10754558 or another polymorphism in linkage disequilibrium with it possibly has an influence on the processing of pro-IL-1β.

Notably, higher levels of IL-1β (Mann–Whitney test, $p < 0.0001$) were observed among patients (mean ± SD: 19.24 ± 3 pg/mL) when compared with controls (mean ± SD: 11.80 ± 1.0 pg/mL) (Fig 1). Our data show that individuals with the C/C genotype have a higher concentration of IL-1β than those with the G allele.

## Discussion

NLRP3 inflammasome is a cytoplasmic sensor that, upon activation, recruits various proteins to form a multiprotein complex, activating caspase-1. Caspase-1 process pro-IL1β into IL-1β cytokine. NLRP3 inflammasome is regulated by caspase recruitment domain 8 (CARD8) [20]. Recently, many studies have evidenced that the NLRP3 inflammasome is involved in the recognition of HCV and processing pro-IL-1β in liver macrophages and also hepatocytes infected with HCV [26–28].

The genetic background of an individual may affect the control, susceptibility and chronicity of HCV infection [29]. A recent study evaluated 201 Egyptian hepatitis C patients and found that the *IL1B* rs1143629 located in intron A/A genotype was prevalent in HCV patients [OR = 1.7 (95% CI = 1–2.8)] [30]. In this study, we performed the rs16944 that is located in the promoter region at position -511 and showed that individuals with the C/C genotype were prevalent among the patients with HCV and have 58% chances higher of developing hepatitis (OR = 1.58 [95% CI 0.88–2.9]) compared to carriers homozygous for the T allele, suggesting

polymorphisms present in the *IL1B* gene may be associated to the development of hepatitis caused by HCV. Comparing the genotypes with circulating plasma levels of IL-1β, did show any influence of the SNP on the plasma levels.

Interestingly, the *IL1B* rs16944 allele C has been identified as a genetic marker for the development of hepatocarcinoma in patients with chronic HBV infection [31]. Furthermore, genome-wide association study in southwest China revealed 6 new loci, including the rs16944 SNP in the *IL1B* gene, associated with an increased prevalence rate of chronic hepatitis B [32]. Altogether, the rs16944 may be a genetic modifier of hepatitis caused by either HBV or HCV.

We observed higher plasma levels of IL1β among the patients with HCV compared to the HD. Similarly in other studies, higher concentrations of plasma circulating cytokines IL-18 and IL-1β were observed in hepatitis B patients [33]. In hepatitis C virus infection, studies show that serum IL-18 and IL-1β concentrations are higher in patients with chronic HCV infection and HCV-related cirrhosis compared with healthy controls [34–36].

NLRP3 rs10754558 polymorphism (3'UTR region) has been studied in several infectious diseases (HIV-1 and HTLV1 infection) [37, 38], neoplastic processes [39], hematological diseases [40], inflammatory and autoimmune disorders [41, 42]. The NLRP3 rs10754558 variant has been correlated with higher mRNA expression by either altering expression of an enhancer activity or mRNA stability [43]. In this study, individuals with the C/C genotype have a higher concentration of IL-1β than those with the G allele.

In one study among patients with Hepatitis C (sustained virological responders against non-responders groups), the NLRP3 rs10754558 had no influence on treatment [30]. We did not observe any differences in alleles or genotypes frequencies between the HD and the patients. However, heterozygosity for NLRP3 rs10754558 seems to associate with severe disease. The loss of heterozygosity among ≥F2, a decrease from 43% among the <F2 compared to 25% among the ≥F2, may suggest that individual with this genotype dies earlier compared to individuals homozygous for GG or CC. Notably, patients ≥F2 are older than the patients <F2. Interestingly, we observed an increase from 5% of the GG genotype among the <F2 to 16% among the ≥F2.

Considering that the various components of inflammasomes interact, we evaluated whether polymorphisms located in different genes could interact and have a combined effect on HCV infection and influence the development of liver fibrosis. Gene-gene interaction analysis was performed between cases and controls, as well as comparisons between groups of patients with mild (<F2) and advanced fibrosis (≥F2).

Gene-gene interaction showed that individuals heterozygotes for both *CARD8* rs2009373 and *IL1B* rs16944 are less prone to hepatitis C development ($p_{adj}$ = 0.039). Similarly, heterozygote carriers for *CTSB* rs1692816 and *AIM2* rs1103577 ($p_{adj}$ = 0.008) or for *IL18* rs187238 and *NLRP3* rs10754558 ($p_{adj}$ = 0.005), have less chances to the development of hepatitis C.

In the analysis of gene interaction between subgroups of mild fibrosis (<F2) and advanced fibrosis (≥F2), individuals homozygous for the T allele of *CARD8* rs2009373 and heterozygous for *IL18* rs187238 ($p_{adj}$ = 0.028), have mild form of fibrosis. Altogether, the approach of gene-gene interactions demonstrates that not only one gene but several genes are involved in the development of hepatitis.

The production of cytokines IL-1β and IL-18, resulting from the activation of inflammasomes, can modulate adaptive immune responses in complex and diverse ways. IL-18 stimulates the production of IFN-γ and the proliferation of Th1 cells. IL-1β can induce the survival and proliferation of naïve T cells, via positive regulation of the IL-2 receptor. Humoral responses can also be increased by IL-1β, directly via increased proliferation of B cells or indirectly by the positive regulation of costimulatory molecules in T cells [42].

Therefore, the activation of the inflammasome can significantly affect an adaptive immune response, which is necessary for the successful elimination of HCV [26, 44]. However, it is necessary to maintain the balance in these immune responses, to avoid cell damage and tissue impairment, as in the case of patients with chronic hepatitis C, who have a disrupted immune response against HCV [45, 46].

This study has some limitations. Our sample size is small and does not allow intra-comparison of the genotype's combination studied with IL-1β cytokine. However, it showed that the combinations of these polymorphisms seem to influence in the chronic hepatic disease. Further studies are needed to confirm this preliminary finding.

## Conclusion

The present study demonstrated an association between inflammasomes genes and development of hepatitis C. We note that individuals homozygotes for the C allele of the *IL1B* C/T rs16944 seem to be susceptible to HCV infection and individuals homozygote for the A allele still have 50% less chances of developing advanced fibrosis. NLRP3 rs10754558 C/C genotype showed an influence on the serum cytokine concentration of IL-1β. Studies of these genes in different world populations should help understand the importance of these variants in the role of host genetic variability in the clinical presentation and development of hepatitis C and fibrosis hepatic. However, further studies are recommended to confirm our findings.

## Contribution to the field statement

Hepatitis C is an inflammatory liver disease caused by the hepatitis C virus (HCV). Nearly 71 million individuals are estimated to be infected worldwide and around 70% will progress to chronic form of the disease. Since nearly two-third of HCV-infected individuals progress to chronic form of the disease while the rest can eliminate the virus or remain asymptomatic, this suggests that the host genetic background may play an important role in the clinical outcome of HCV infection.

NLRP3 inflammasome is a cytoplasmic sensor that, upon activation, recruits various proteins to form a multiprotein complex, activating caspase-1. Caspase-1 process pro-IL1β into IL-1β cytokine. Recently, many studies have shown that the NLRP3 inflammasome is involved in the recognition of HCV and processing pro-IL-1β in hepatic macrophages and hepatocytes infected with HCV. The production of cytokines IL-1β and IL-18, resulting from the activation of inflammasomes, can modulate immune responses in complex and diverse ways.

We believe that genetic background of an individual, related to inflammasome complex proteins, may affect the control, susceptibility and chronicity of HCV infection, possibly because it influences the magnitude of the antiviral immune response and inflammation during infection, with consequences for the patient's clinical presentation.

## Supporting information

**S1 Fig. Serum concentration of IL-1β cytokine in 151 patients with hepatitis C, stratified according to the degree of hepatic fibrosis, analyzed as a function of the *IL1β* (rs16944), *IL-18* (rs187238), *CARD8* (rs2009373), *CTSB* (rs1692816), *AIM2* (rs1103577) polymorphisms.** Data are expressed as mean ± standard deviation of circulating concentration (pg/mL) of IL-1β cytokine. Statistical analyzes were performed by ANOVA (nonparametric analysis of variance), with Kruskal-Wallis test, followed by Dunns post-test to compare pairs.
(TIF)

**S2 Fig. Serum concentration of IL-1β cytokine of 49 patients with hepatitis C, without treatment, stratified according to the degree of hepatic fibrosis, analyzed as a function of the *IL1β* (rs16944), *IL-18* (rs187238), *CARD8* (rs2009373), *CTSB* (rs1692816), *AIM2* (rs1103577) polymorphisms.** Data are expressed as mean ± standard deviation of circulating concentration (pg/mL) of IL-1β cytokine. Statistical analyzes were performed by ANOVA (nonparametric analysis of variance), with Kruskal-Wallis test, followed by Dunns post-test to compare pairs.
(TIF)

**S1 Table. Sequences of the probes used for Real Time PCR genotyping.**
(DOCX)

**S2 Table. Primers and PCR conditions for the studied polymorphisms.**
(DOCX)

## Acknowledgments

We acknowledge the collaboration of Ambulatório de Hepatopatia da Fundação de Medicina Tropical -Doutor Heitor Vieira Dourado (FMT-HVD) and laboratory support of the Laboratório de Genômica da Fundação Hospitalar de Hematologia e Hemoterapia do Amazonas (HEMOAM), Laboratório de Micologia (INPA) and Laboratório de Imunologia Celular (UFAM).

## Author Contributions

**Conceptualization:** Diana Mota Toro, Marilú Victória Barbieri, Aya Sadahiro, Andréa Monteiro Tarragô, Adriana Malheiro.

**Data curation:** Diana Mota Toro, Nadja Pinto Garcia, Allyson Guimarães Costa, Aya Sadahiro.

**Formal analysis:** Diana Mota Toro, Rajendranath Ramasawmy, Nadja Pinto Garcia, Flamir Silva Victória, Eduardo Antônio Donadi, Allyson Guimarães Costa, Mauricio Morishi Ogusku, Aya Sadahiro, Andréa Monteiro Tarragô.

**Funding acquisition:** Marilú Victória Barbieri, Allyson Guimarães Costa, Aya Sadahiro, Adriana Malheiro.

**Investigation:** Pedro Vieira Silva Neto, Grenda Leite Pereira, Priscila Santos Sarmento, Hanna Lara Silva Negreiros Dray, Keyla Santos Sousa, Juliana Santos Affonso, Marilú Victória Barbieri, Flamir Silva Victória, Eduardo Antônio Donadi, Mauricio Morishi Ogusku, Aya Sadahiro, Andréa Monteiro Tarragô, Adriana Malheiro.

**Methodology:** Diana Mota Toro, Rajendranath Ramasawmy, Pedro Vieira Silva Neto, Grenda Leite Pereira, Priscila Santos Sarmento, Hanna Lara Silva Negreiros Dray, Keyla Santos Sousa, Juliana Santos Affonso, Jéssica Albuquerque Silva, Nadja Pinto Garcia, Marilú Victória Barbieri, Flamir Silva Victória, Eduardo Antônio Donadi, Mauricio Morishi Ogusku, Aya Sadahiro, Andréa Monteiro Tarragô, Adriana Malheiro.

**Supervision:** Andréa Monteiro Tarragô.

**Writing – original draft:** Diana Mota Toro, Allyson Guimarães Costa, Andréa Monteiro Tarragô, Adriana Malheiro.

**Writing – review & editing:** Rajendranath Ramasawmy, Marilú Victória Barbieri, Eduardo Antônio Donadi.

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
