## [Decision Letter · Decision Letter 0]

23 Feb 2021

PONE-D-20-35265

Inflammasome genes polymorphisms may influence the development of Hepatitis C in the Amazonas, Brazil

PLOS ONE

Dear Dr. Costa,

Thank you for submitting your manuscript to PLOS ONE. After careful consideration, we feel that it has merit but does not fully meet PLOS ONE’s publication criteria as it currently stands. Therefore, we invite you to submit a revised version of the manuscript that addresses the points raised during the review process.

We look forward to receiving your revised manuscript.

Kind regards,

Narasimha Reddy Parine, Ph.D

Academic Editor

PLOS ONE

Journal Requirements:

Reviewers' comments:

Reviewer's Responses to Questions

**Comments to the Author**

1. Is the manuscript technically sound, and do the data support the conclusions?

Reviewer #1: Yes

Reviewer #2: Yes

2. Has the statistical analysis been performed appropriately and rigorously? 

Reviewer #1: Yes

Reviewer #2: Yes

3. Have the authors made all data underlying the findings in their manuscript fully available?

Reviewer #1: Yes

Reviewer #2: Yes

4. Is the manuscript presented in an intelligible fashion and written in standard English?

Reviewer #1: Yes

Reviewer #2: Yes

5. Review Comments to the Author

Reviewer #1: Article Report

Title: Inflammasome genes polymorphisms may influence the development of Hepatitis C in

the Amazonas, Brazil

Summary of article: Diana Mota Toro et al, from Programa de Pós-Graduação em Imunologia Básica e Aplicada, Universidade Federal do Amazonas, Manaus, Amazonas, Brazil. The main focus of this research work is to Hepatitis C and associated polymorphisms of inflammasomes. Author investigated 151 patients with chronic hepatitis C and 206 healthy blood donors’ individuals (HD).

Polymorphisms in the IL1B and IL18 genes were genotyped by PCR-RFLP, while

NLRP3, CARD8, CTSB and AIM2 by RT- PCR.

Serum assay of IL-1β cytokine was performed by ELISA. 84 patients presented mild fibrosis (<f2) 67="" advanced="" and="" fibrosis="">Comment: Minor revision

The work is very important in terms of better understanding of association between inflammasomes genes and development of hepatitis C. However, below comments should be address in main content with references to make it more informative and attractive for wide users.

1. Why author choose PCR-RFLP and RT-PCR for polymorphism study. Reason to use PCR-RFLP should be mentioned in main text.

2. Progression of disease depends on multigenic expression of genes in pathway. Why author choose only 6 genes for this study.

3. Genetic background of populations also responsible for false association in the study. Cast and categories along with regional influence might be responsible for various results. Author justify the concern of population stratification if data available about case and control subjects.

4. Sample size is one of challenges in case-control study, however mild vs advanced fibrosis and combined data should be evaluated with statistical corrections for correct p value. Author explain the severity of disease and other gene role in discussion part.

5. How heterogeneity and homogeneity influence the expression of gene and thus the varied clinical out come in HCV.

6. Is there any other study of similar gene and region population in HCV, Kindly mention latest reference to make it for more better understanding.</f2)>

Reviewer #2: The manuscript entitled “Inflammasome genes polymorphisms may influence the development of Hepatitis C in the Amazonas, Brazil” by Toro et al. sheds some light on the role of the inflammasome genes variations in the in the progression of the chronic hepatitis C in Brazilian patients.

The study is very well designed and well written. I have very few comments.

1. Line 183-185: please adjust the sentence.

2. Table 3 and table 4: Please show the number of patients in each sub-group because I think the numbers are so small especially in table 4. So, the authors should be very cautious when they draw conclusions from these tables, and they should state this clearly in the study.

3. Line 330-340: these two paragraphs are redundant with the results so please try to support your finding with possible explanation or supporting studies if any.

4. Also, in the discussion, it will be great if the authors explain how the findings of this study could help managing the HCV patients in the future.

6. PLOS authors have the option to publish the peer review history of their article (what does this mean?). If published, this will include your full peer review and any attached files.

Reviewer #1: **Yes: **Dr. Anshuman Mishra

Reviewer #2: No

---

## [Author Response · Author response to Decision Letter 0]

15 Apr 2021

Manaus, April 08th, 2021

To: Narasimha Reddy Parine, Ph.D.

Academic Editor

Plos One

PONE-D-20-35265

Dear Editor,

We very much appreciate the kind consideration given to our manuscript (MS) PONE-D-20-35265, entitled "Inflammasome genes polymorphisms may influence the development of Hepatitis C in the Amazonas, Brazil". We hope that the replies to the reviewer’s comments will have satisfactorily improved the MS.

Below, we present all the queries made by the reviewers. The changes requested are clearly outlined in the revised manuscript and marked in yellow. We have prepared a list of answers to the reviewers’ comments, which are highlighted in “bold italic”. In the response to each query, we are also including the modified part, as it is in the revised manuscript.

Journal Requirements:

We acknowledge the comment of Plos One Team and inform that Journal Requirements were revised. 

We acknowledge the comment of Plos One Team and inform that Journal Requirements were revised. 

We acknowledge the comment of Plos One Team and inform that Journal Requirements were revised. 

Reviewers' comments:

Reviewer reports:

Reviewer #1:

Title: Inflammasome genes polymorphisms may influence the development of Hepatitis C in the Amazonas, Brazil

Summary of article: Diana Mota Toro et al, from Programa de Pós-Graduação em Imunologia Básica e Aplicada, Universidade Federal do Amazonas, Manaus, Amazonas, Brazil. The main focus of this research work is to Hepatitis C and associated polymorphisms of inflammasomes. Author investigated 151 patients with chronic hepatitis C and 206 healthy blood donors’ individuals (HD).

Polymorphisms in the IL1B and IL18 genes were genotyped by PCR-RFLP, while

NLRP3, CARD8, CTSB and AIM2 by RT- PCR.

Serum assay of IL-1β cytokine was performed by ELISA. 84 patients presented mild fibrosis (Comment: Minor revision

The work is very important in terms of better understanding of association between inflammasomes genes and development of hepatitis C. However, below comments should be address in main content with references to make it more informative and attractive for wide users.

We acknowledge the encouraging comment of Reviewer #1, which clearly indicates that our message was well understood.

1. Why author choose PCR-RFLP and RT-PCR for polymorphism study. Reason to use PCR-RFLP should be mentioned in main text.

We do understand your worry about using two different techniques for alleles discrimination. We used PCR-RFLP for alleles discrimination for IL1B rs16944 and IL18 rs187238 as the technique is very well established in the literature and is less expensive than RT-PCR using TaqMan probes on demand. However, we have included in the text Line 147-152 for readers to understand the technique PCR-RFLP.

2. Progression of disease depends on multigenic expression of genes in pathway. Why author choose only 6 genes for this study.

You are right to point that disease progression depends on multigenic expression of genes in pathway. In this study we chose these genes because they are the most studied in various inflammatory diseases. However, ongoing research in our laboratory to decipher the immunogenetics mechanism involved in the development of the disease is pursuing.

3. Genetic background of populations also responsible for false association in the study. Cast and categories along with regional influence might be responsible for various results. Authors justify the concern of population stratification if data available about case and control subjects.

Thank you for reminding us about spurious association due to genetics background. Cases and controls were carefully selected to avoid genetic heterogeneity. Furthermore, all the participants are unrelated individuals to avoid bias association.

4. Sample size is one of challenges in case-control study, however mild vs advanced fibrosis and combined data should be evaluated with statistical corrections for correct p value. Authors explain the severity of disease and other gene role in discussion part.

We agree with you that sample size is very important in genetics association studies especially when stratifications into different groups are performed. Conscious of this, we have adjusted all the p values for sex and age. We did not performed corrections of p value for multiple comparisons as we think that Bonferroni corrections is too conservative. Furthermore, we think that our study can be a lead for other groups to replicate or not these associations in different populations. We pointed these limitations in the discussion section (line357-361).

5. How heterogeneity and homogeneity influence the expression of gene and thus the varied clinical outcome in HCV.

We believe that genetic heterogeneity or homogeneity may the same influence on gene expression in a pathway because we all have the same genes. However, we agree that environmental factors may play a role. Furthermore, evolution may have shaped genetics substructure as we often observed different population allele frequency for a given variant. That is why we always select our population of study from the same genetic background. It might be philosophical today in post-genomic era as always race was attributed to it.

6. Is there any other study of similar gene and region population in HCV, Kindly mention latest reference to make it for more better understanding.

To the best of our knowledge, we believed to have referred most of articles pertinent to or study. Finally, we appreciate very much your positive comments to help us improve our manuscript.

Reviewer #2: 

The manuscript entitled “Inflammasome genes polymorphisms may influence the development of Hepatitis C in the Amazonas, Brazil” by Toro et al. sheds some light on the role of the inflammasome genes variations in the in the progression of the chronic hepatitis C in Brazilian patients.

The study is very well designed and well written. I have very few comments.

We acknowledge the encouraging comment of Reviewer #2, which clearly indicates that our message was well understood.

1. Line 183-185: please adjust the sentence.

 We have improved the sentence to make it understandable.

2. Table 3 and table 4: Please show the number of patients in each sub-group because I think the numbers are so small especially in table 4. So, the authors should be very cautious when they draw conclusions from these tables, and they should state this clearly in the study.

We did state this in the limitations of the study.

3. Line 330-340: these two paragraphs are redundant with the results so please try to support your finding with possible explanation or supporting studies if any.

We corrected in the text.

4. Also, in the discussion, it will be great if the authors explain how the findings of this study could help managing the HCV patients in the future.

We included a paragraph about this. Line 361-364.

We believed that the new changes have significantly improved the quality of our manuscript. We would like to thank Plos One members and reviewers for their dedication in providing their valuable and interesting comments on this article. 

We sincerely hope that the revised version of our manuscript meets the high standards of Plos One publications and is, therefore, acceptable for publication in this journal.

With best wishes,

Allyson Guimarães da Costa (on behalf of all authors)

---

## [Decision Letter · Decision Letter 1]

7 Jun 2021

Inflammasome genes polymorphisms may influence the development of Hepatitis C in the Amazonas, Brazil

PONE-D-20-35265R1

Dear Dr. Costa,

We’re pleased to inform you that your manuscript has been judged scientifically suitable for publication and will be formally accepted for publication once it meets all outstanding technical requirements.

Kind regards,

Cinzia Ciccacci

Academic Editor

PLOS ONE

Additional Editor Comments (optional):

Reviewers' comments:

Reviewer's Responses to Questions

**Comments to the Author**

1. If the authors have adequately addressed your comments raised in a previous round of review and you feel that this manuscript is now acceptable for publication, you may indicate that here to bypass the “Comments to the Author” section, enter your conflict of interest statement in the “Confidential to Editor” section, and submit your "Accept" recommendation.

Reviewer #1: All comments have been addressed

2. Is the manuscript technically sound, and do the data support the conclusions?

Reviewer #1: Yes

3. Has the statistical analysis been performed appropriately and rigorously? 

Reviewer #1: Yes

4. Have the authors made all data underlying the findings in their manuscript fully available?

Reviewer #1: Yes

5. Is the manuscript presented in an intelligible fashion and written in standard English?

Reviewer #1: Yes

6. Review Comments to the Author

Reviewer #1: Accepted with selected respons recitation in the main text. Perticularly querry no. 2 to 4 is important.

7. PLOS authors have the option to publish the peer review history of their article (what does this mean?). If published, this will include your full peer review and any attached files.

Reviewer #1: **Yes: **Dr. Anshuman Mishra

---

## [Editor Report · Acceptance letter]

15 Jun 2021

PONE-D-20-35265R1 

Inflammasome genes polymorphisms may influence the development of Hepatitis C in the Amazonas, Brazil 

Dear Dr. Costa:

I'm pleased to inform you that your manuscript has been deemed suitable for publication in PLOS ONE. Congratulations! Your manuscript is now with our production department. 

Kind regards, 

on behalf of

Dr. Cinzia Ciccacci 

Academic Editor

PLOS ONE